# Thinking Temporal Automatic White Balance: Datasets, Models and Benchmarks

## ABSTRACT

Temporal Automatic White Balance (TAWB) corrects the color cast within each frame, while ensuring consistent illumination across consecutive frames. Unlike conventional AWB, there has been limited research conducted on TAWB for an extended period. However, the growing popularity of short-form videos has increased focus on video color experiences. To further advance research on TAWB, we aim to address the bottlenecks associated with datasets, models, and benchmarks. 1) Dataset challenge: Currently, only one TAWB dataset (BCC), captured with a single camera, is available. It lacks temporal continuity due to challenges in capturing realistic illuminations and dynamic raw data. In response, we meticulously designed an acquisition strategy based on the actual distribution pattern of illuminations and created a comprehensive TAWB dataset named CTA comprising 6 cameras that offer 12K continuous illuminations. Furthermore, we employed video frame interpolation techniques, extending the captured static raw data into dynamic form and ensuring continuous illumination. 2) Model challenge: Among the two prevailing TAWB methods, both rely on LSTM. However, the fixed gating mechanism of LSTM often fails to adapt to varying content or illuminations, resulting in unstable illumination estimation. In response, we propose CTANet, which integrates cross-frame attention and RepViT for self-adjustment to content and illumination variations. Additionally, the mobile-friendly design of RepViT enhances the portability of CTANet. 3) Benchmark challenge: Currently, there is no benchmark of TAWB methods on illumination and camera types to date. Addressing this, a benchmark has been proposed by conducting a comparative analysis of 8 cutting-edge AWB and TAWB methods with CTANet across 3 typical illumination scenes and 7 cameras from two representative datasets. Our dataset and code are available in supplementary material.

## KEYWORDS

Temporal automatic white balance, Benchmark and dataset, Low-level

## 1 INTRODUCTION

Temporal Automatic White Balance (TAWB) is a crucial process that corrects the color cast within each frame while maintaining

**Unpublished working draft. Not for distribution.**

Permission to make digital or hard copies of all or part of this work for personal or classroom use is granted without fee provided that copies are not made or distributed for profit or commercial advantage and that copies bear this notice and the full citation on the first page. Copyrights for components of this work owned by others than the author(s) must be honored. Abstracting with credit is permitted. To copy otherwise, or republish, to post on servers or to redistribute to lists, requires prior specific permission and/or a fee. Request permissions from permissions@acm.org.

*ACM MM, 2024, Melbourne, Australia*

© 2024 Copyright held by the owner/author(s). Publication rights licensed to ACM.
ACM ISBN 978-x-xxxx-xxxx-x/YY/MM
https://doi.org/10.1145/nnnnnnn.nnnnnnn

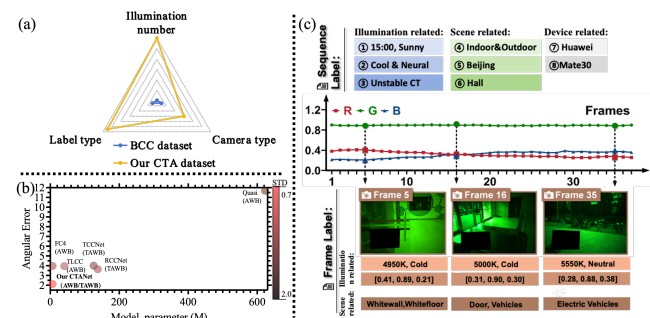

**Figure 1: (a) and (c): Our CTA dataset addresses the issues faced by the TAWB dataset in terms of illumination numbers, continuity, and camera numbers. (b): our CTANet handles the illumination continuity problem of the TAWB methods by incorporating cross-frame attention and RepViT, achieving the best illumination estimation accuracy and stabilization (lowest average angular error and STD error) with the lightest parameters .**

illumination continuity across successive frames. There are *significantly fewer* studies related to TAWB [19, 18, 8, 17, 30] as compared to traditional AWB studies that are specific to still images [9, 17, 22, 3, 12, 13, 5, 28, 2, 11].

TAWB research has become extremely important since mobiles and cameras are increasingly used for video acquisition and reproduction [8]: In this context, the discomfort of poor illuminant correction is potentially amplified if such correction also changes over time without justification, thus introducing unpleasant flickering artifacts [4, 33, 27, 29]. However, TAWB research encounters notable challenges, particularly regarding datasets, methods, and benchmark.

According to Imaging Model [12], real-world illuminations and cameras significantly influence the perception of illumination colors in the raw-RGB space, which serve as the ground truth (GT) for TAWB. Real-world illuminations are primarily natural or artificial, each with distinct color temperature ranges. Including these broadens the illumination range in raw RGB spaces for the TAWB datasets. Additionally, cameras with different image signal processors (ISPs) or sensors impact the raw illumination distributions, even under identical real color temperatures, increasing the distribution diversity in raw spaces [20]. However, as in Fig.1(a), current available TAWB datasets, BCC [18], do not adequately cover real-world illuminations and lack camera diversity. Other datasets by Cirurea [10], Prinet *et.al* [17] and Yoo *et.al* [30] are rarely utilized due to their limited data format[10] and application scope[17, 30]. Due to the difficulties in obtaining raw-format video, the TAWB dataset, BCC[18], has chosen to capture continuous raw frames to mimic video content. Although this strategy simplifies the process

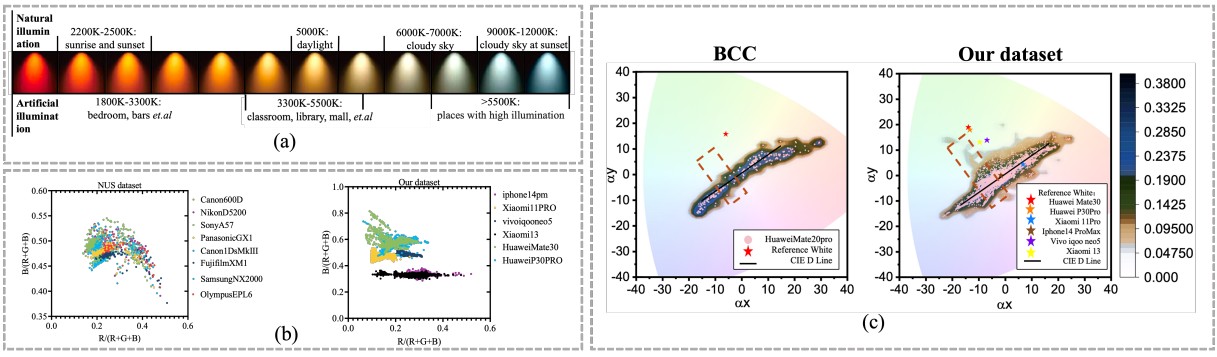

Figure 2: (a) Common CTs for natural and artificial illuminations. (b) Raw illumination colors for different cameras in the NUS dataset [9] and our CTA dataset. (c) Illumination distributions in the Angle-Retaining Chromaticity diagram [6] for Gray Ball dataset, BCC dataset and our CTA dataset, normalized by sensor-specific reference whites.

of capturing video sequences, it results in the loss of crucial illumination continuity information. As a result, models trained on such datasets may struggle to handle temporal variations in real-world videos.

TAWB methods should maintain stable illumination to prevent flickering in the corrected video. This necessitates flexible adaptation to temporal changes in input frames, such as changes in illuminations or contents, to use shared features for illumination estimation. However, both available methods, RCCNet[19] and TC-CNet[18], use the Long Short-Term Memory (LSTM)[21] to gather temporal features. The gating mechanism is a crucial component of LSTM, but its parameters are fixed. This makes it difficult for existing methods to adjust to changes in contents or illuminations of input frames. As a result, the extracted temporal features cannot reflect the shared information between different frames, which ultimately results in unstable illumination estimation.

To comprehensively advance TAWB research, we make two contributions. First, we build a large-scale dataset called **C**omprehensive **T**emporal **A**WB (CTA), encompassing 12637 real-world continuity illuminations (natural, artificial, and mixed), six popular mobile cameras with different ISPs and sensors, and 11 detailed annotations for each sequence (Fig.1(a) and (c)). Specifically, 1) We meticulously designed an acquisition strategy guided by real illumination pattern [**2004:ITE:1009386.1010128**]. Our aim is to ensure that the illuminations we collect cover a wide range of real-world situations, thus, providing a comprehensive representation of the illumination diversity. 2) To address the difficulty of capturing dynamic raw data, we implemented a video interpolation process. This step allows for the continuous capture of static raw data and expands it into a dynamic form, thus guaranteeing illumination continuity.

Second, we propose a TAWB method called CTANet to maintain the temporal continuity of the estimated illumination. Unlike the LSTM with fixed weights employed in existing methods, we utilize cross-frame attention to extend RepViT [25], a lightweight attention network, to the temporal dimension. Specifically, we design the Spatial-Temporal Stage (ST-Stage) block, which utilizes a combination of the token mixer and channel mixer to extract intra-frame spatial and channel features. Additionally, the block incorporates cross-frame attention to identify and combine similar

features from preceding frames to the target frame, forming shared features. This strategy allows our model to be adaptive to changes in input content and illumination, ensuring the temporal continuity of the estimated illuminations (lowest STD in Fig.1(b)).

We perform a thorough benchmark analysis of eight cutting-edge AWB and TAWB methods with CTANet across three typical illumination scenes and seven cameras based on the BCC dataset and our CTA dataset. Our CTANet excels in correction accuracy and temporal consistency of estimated illuminations, surpassing existing TAWB methods with the least number of parameters (Fig.1). This analysis also points out some possible challenges for future research, such as the accuracy and stability of solid color scenes, cross-camera generalization, *etc*. We hope that our work can advance the field of TAWB.

## 2 CHALLENGES AND SUGGESTIONS IN TAWB

### 2.1 TAWB Dataset

***Challenge 1***: *The current datasets are insufficient in capturing the diverse range of realistic illumination types.*

Natural illumination and artificial illumination are two major types of realistic illumination. However, as illustrated in Fig.2(c), existing available TAWB dataset, BCC [18], shows limited coverage of typical natural illuminations (CIE series D illuminations, represented by the black lines) and inadequate artificial illuminations, especially those between the greenish and magentaish directions (indicated by the red boxes). This gap is crucial, as these illuminations are vital to understanding human perception of illuminations [7]. Additionally, these datasets considerably lacked mixed illuminations that are common in scene transition sequences.

***Suggestion 1***: *Considering the distinct color temperature patterns of real-world illuminations [15], accurately capturing them requires attention to key factors: 1) the time of day, weather conditions and geographical locations for natural illuminations; and 2). the specific function of the indoor space for artificial illuminations. Refer to Fig.2(a).*

***Challenge 2***: *The current datasets are limited in camera diversity, a critical factor for ensuring the comprehensiveness.*

Camera diversity is crucial for TAWB research since it increases the diversity of raw illumination distributions. For example, due

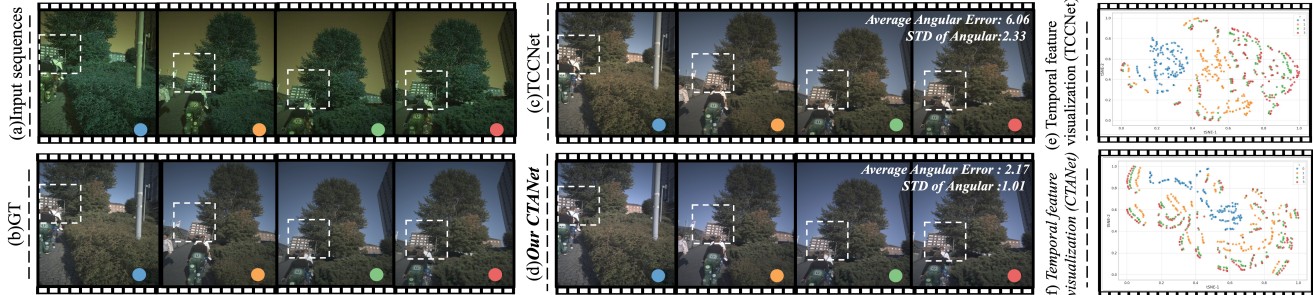

**Figure 3: Visualization of unstable illumination estimation with existing methods(TCCNet for example).**

to inherent ISP differences in DSLR cameras, their raw illumination color distributions vary significantly, resembling a rainbow as depicted in Fig. 2(b)(NUS AWB dataset[9]). This variation is further amplified in Fig. 2(c) by incorporating sensor differences from various mobile cameras. However, the BCC dataset predominantly used a single camera type, which restricts the diversity of raw illumination color distributions. This also hindered the support of multi-camera generalization, a crucial aspect for TAWB [1].
***Suggestion 2:***1) *Employ cameras with varying ISPs or sensors to ensure diversity. 2) For multi-camera generalization, ensure that adopted sequences from different cameras contain similar content.*

## 2.2 TAWB Baseline Method

***Challenge 3:*** *TAWB methods should maintain illumination continuity to prevent flickering in the corrected video. However, the fixed gating mechanism of LSTM makes existing methods fail to adapt to varying contents or illuminations, leading to unstable illumination estimation.*

Assuming $I_t$ as the target frame, $\{I_{t-(N-1)} \cdots I_{t-1}\}$ are $N$ preceding frames, existing methods [19, 18] are formed as:

$$\hat{l}_t = \mathcal{P}(\mathcal{T}(\mathcal{S}(\{I_t\}^N))), \qquad (1)$$

where $\mathcal{S}(\cdot)$ is the spatial feature extractor that extracts features from input frames separately. The temporal feature extractor $\mathcal{T}(\cdot)$ accepts these features and outputs the temporal feature $x_t^{te}$ for the target frame $I_t$, which are then used for estimating illumination $\hat{l}_t$ via the prediction head $\mathcal{P}(\cdot)$.

Compared to AWB methods, TAWB methods further consider the temporal correlation of illuminations, to ensure that the inter-frame changes are smooth rather than sudden jumps. This requires that the temporal feature $x_t^{te}$ can represent the shared information of all input frames, *e.g.*, buildings or trees appearing in these frames to build the stable illumination estimation cues.

However, existing methods (RCCNet [19] and TCCNet [18]) both utilize the gating mechanism of LSTM (forgetting and memory gates) to aggregate the temporal feature $x_t^{te}$. Since the weights of this mechanism are fixed after training, it leads to the fact that existing methods cannot always extract the shared features of the input frames, negatively affecting the sequences with large content or illumination change. As Fig.3(e) shows, when the input frames undergo content change (Fig.3(a)), the temporal feature for the target frame ($x_t^{te}$, red scatters) is far from preceding frames, especially

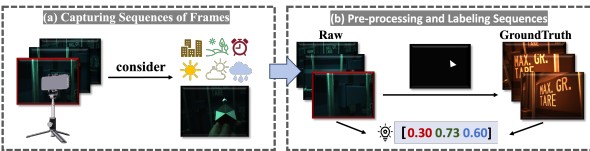

**(a) Existing datasets**

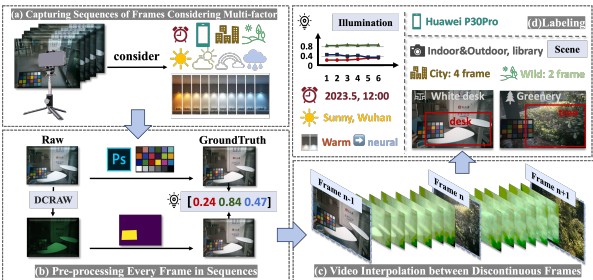

**(b) Our CTA dataset**

**Figure 4: Data construction process comparison.**

the first frame (blue scatters). Inevitably, in Fig.3(c) the error is large (Average) and unstable (STD).
***Suggestion 3:*** *Instead of the fixed gather mechanism, temporal feature extraction should adapt to content and illumination changes.*

## 3 PROPOSED DATASET

Comprehensive datasets are instrumental in advancing TAWB research. However, as discussed in Sec.2.1, there is only one available TAWB dataset (BCC) captured using a single camera, which lacks temporal continuity due to the challenges in capturing realistic illuminations and dynamic raw data. To address this problem, our data collection incorporates multiple factors related to illuminations and cameras in accordance with ***Suggestion 1*** and ***Suggesting 2***. Moreover, the video frame interpolation (VFI) step is applied to enhance the illumination temporal continuity. Details are as follows.

## 3.1 Dataset Acquisition

As discussed in Sec.2.1, there are two challenges for TAWB datasets, here we describe how to address them to acquire the CTA dataset. For ***Challenge 1***, we followed the real illumination patterns in ***Suggestion 1*** to capture natural and artificial illuminations. To

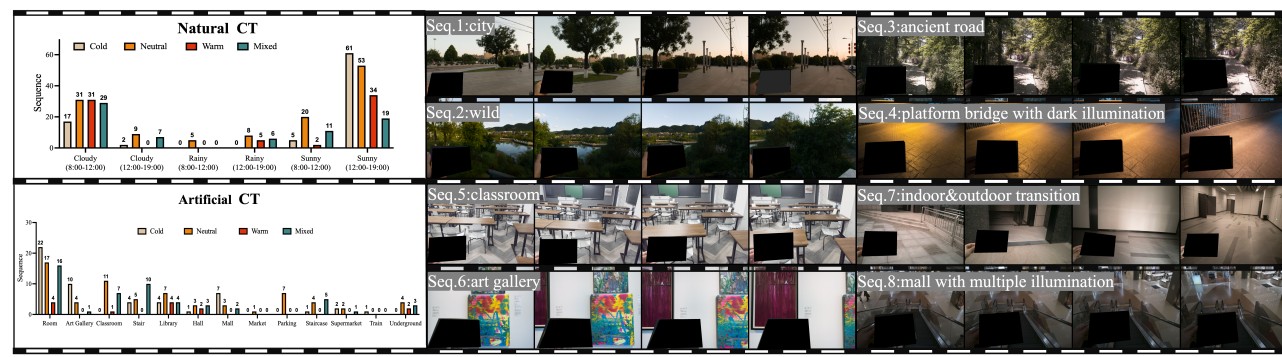

**Figure 5: Representative scenes and corresponding color temperature from our dataset. (Note that the keyframes are visualized for complete sequence content and processed by Dcraw into a JPEG format for better display (without AWB).)**

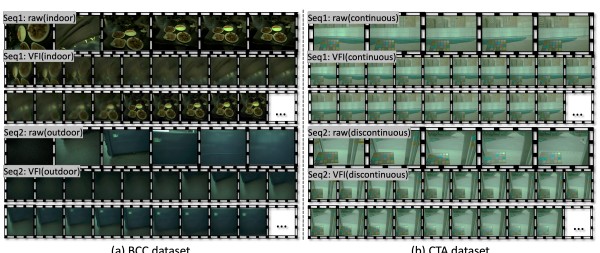

**Figure 6: Raw sequences with VFI sequences of BCC and CTA dataset(brightness adjusted for ease of viewing)**

accurately represent varying color temperatures in natural illuminations, we recorded sequences at different moments from 7:00 AM to 8:00 PM, including three typical weather conditions - sunny, cloudy and rainy - across 334 outdoor scenes encompassing both city and wild scenes. To encompass the unique color temperature preferences of indoor artificial illumination, we included 231 varied indoor settings such as offices, classrooms, libraries, malls, and art galleries. The statistics of illumination color temperatures and examples are in Fig. 5. 75 transition sequences between scenes were also consciously captured to better approximate real-life video shooting.

In Fig.2(c), the captured illuminations completely cover the CIE series D illuminants (black line), which indicates its representations of natural illuminations. Notably, the artificial types between the greenish and magentaish directions (red box) are also included, which is critical for color constancy while current datasets don't have. Overall, the CTA dataset surpasses existing datasets in providing a broad spectrum of real-world illuminations, both natural and artificial. This addresses *Challenge 1*, further facilitates the development of TAWB research.

Following *Suggestion 2*, we implemented two strategies to address *Challenge 2*. First, we selected six widely-used mobile cameras for the data collection. These included Huawei P30Pro and Huawei Mate30, iPhone 14ProMax, Xiaomi 11Pro, Xiaomi 13, and Vivo iqooneo5. The distinctive RYYB sensors of the Huawei models, in contrast to the RGGB sensors in the other mobile cameras, add to the technological diversity of our dataset. Secondly, for effective multi-camera benchmarking, we carefully captured and selected

the sequences with common objects in different scenes, such as trees and streets for outdoor scenes and furniture for indoor settings. All cameras were set to capture pure raw data, with automatic enhancements like HDR turned off.

In Figure 2(b), the raw illumination distributions from the six selected mobile cameras exhibit distinct characteristics. This variation is largely attributed to the inherent differences in their ISPs and sensors[23]. These results demonstrate the effectiveness of our strategies in addressing *Challenge 2*, as per *Suggestion 2*.

## 3.2 Video Frame Interpolation

After acquisition, we enhanced the illumination continuity of captured static raw data by performing the VFI strategy after data acquisition. Existing datasets ignored the illumination discontinuity problem. For example, the only available dataset, BCC, assumed that the temporal illumination in the sequence is constant, and directly extracts the illumination color of the first frame as that of the sequence (Fig.4a). However, we found that there exist scene transitions in most of the sequences, which were often accompanied by illumination continuous changes, as shown in Fig.6(a). To solve this problem, we implemented two specific strategies as in Fig.4b. Our first strategy involved placing a ColorChecker in every scene, ensuring its visibility in each frame. This setup enabled us to accurately record the changes in temporal illumination colors, in line with the extraction process detailed in [1]. Secondly, we add a VFI technique, EMA-VFI [31] to smooth illumination transitions. This technique interpolates frames by extracting content motion features and appearance color features highly correlated with illuminations. Specifically, we set the interpolation time step between two frames to 8 to get a video frame rate of 24fps. Examples of interpolated frames in BCC and our CTA dataset are shown in Fig. 6. Our datasets will be available in both frame-interpolated and frame-uninterpolated sequences.

## 3.3 Data Annotation

Comprehensive annotation of real-world illuminations and cameras will not only help to better understand their impact on TAWB research, but also contribute to the multimodal development of TAWB. However, existing datasets contained only illuminations (GT). To improve this, we annotated illuminations and cameras

**Figure 7: Network structure of our CTANet. CTANet consists of two components: an ST feature extractor to extract the temporal features that can represent the shared features of the input frames and an illumination prediction head to output the illumination colors of the target frame.**

based on their associated factors in **Suggestion 1 and 2**. In addition, to characterize the scene content, we labeled the contents by object detection method[26] and manually provided color descriptions. An annotation example is in Fig.1(c), there are 11 descriptions for each sequence, including temporal raw illumination colors, illumination color temperature types, scene type, content, colors, location, capturing time, weather conditions, the brand and model of applied mobile cameras.

## 4 PROPOSED METHOD

As discussed in Sec.2.2, due to the fixed gating mechanism of LSTM, two existing TAWB methods, RCCNet and TCCNet, suffer from the unstable illumination estimation problem, especially for the sequences with varying contents or illuminations.

To address this problem, we proposed the CTANet. Unlike existing methods that connect spatial and temporal feature extractors in tandem (Eq.1), our CTANet integrated spatial feature extraction and temporal feature extraction:

$$\hat{l}_t = \mathcal{P}(\mathcal{ST}(\{I_t\}^N)), \tag{2}$$

where $\mathcal{ST}(\cdot)$ is the spatial-temporal (ST) feature extractor with four stages. In each stage, two ST-Stage blocks are set to gather temporal features. The setup of the illumination predictor head is the same as RCCNet and TCCNet. In the following, we introduce the ST feature extractor and ST-Stage block separately.

### 4.1 Spatial-temporal Feature Extractor

Our goal in constructing the ST feature extractor is to extract the temporal features that can represent the shared information of input frames. According to **Suggestion 3**, this requires the ST feature extractor to adapt to content and illumination changes. Benefiting from the design of the self-attention-like mechanism[32, 16], RepViT [25] can partially fulfill this requirement by providing flexibility in spatial feature extraction. In addition, RepViT's mobile-friendly parameter count enhances CTANet's portability. However, RepViT can only extract the spatial features and fails to establish the temporal relationship between frames. To extend the perspective of RepViT to the temporal dimension, we propose to integrate cross-frame attention mechanisms with RepViTBlock to form the ST-Stage block.

As shown in Fig.7, the ST feature extractor consists of a Stem block, four Down Sample blocks, and four Spatial-temporal Stage (ST-Stage) blocks, among which the Stem block and Down Sample block are used to output feature maps at different scales, the ST-Stage block aims to extract temporal features that can represent the shared information of input frames. In the following, we describe the ST-Stage block in detail.

### 4.2 Spatial-temporal Stage Block

As in the blue part of Fig.7, the main component of the ST-Stage block is Spatial-temporal (ST) Attention, which aims to extract the spatial features for input frames flexibly and then find features from the previous frames that are similar to the features in the target frame and fuse them together.

*4.2.1 Spatial-temporal Attention.* The structure of ST attention is in the red block of Fig.7. Support the features of $I_t$ in $k^{th}$ stage as $x_t^k$, the computation of ST attention is:

$$x_t'^k = Conv(Cm(Cross(Tm([x_t^k]))) + Cm(Cross(Tm([x_t^k])). \tag{3}$$

where $[\cdot]$ concatenates the input features from different frames in the $0^{th}$ dimension, the token mixer $Tm(\cdot)$ extracts the spatial features from these inputs separately. Then cross-frame attention $Cross(\cdot)$ finds features in different frames that are similar to the features of the target frame and combines them to gather temporal features, channel mixer $Cm(\cdot)$ increases their channel interactivity, and finally these features are re-weighted by the form of $Conv(\cdot)$ and additive calculation. The above steps aim to fully explore the intra- and inter-frame relationships of the input frames, motivating the extracted temporal features to be representative of the information shared between the input frames. Next is the details of cross-attention.

*4.2.2 Cross-frame Attention.* The structure of cross-attention is in the green block of Fig.7. In detail, we use the feature maps of the preceding frames as the references (Key $K$ and Value $V$) and the target frame as the anchor (Queries $Q$) to constrain the temporal feature extraction of the target frame by exploring the similarity between the target frame and the preceding frames:

$$Cross(Q, K, V) = Softmax(\frac{QK^T}{\sqrt{d}}), \tag{4}$$

| Group | | Train | Val | Test (N) | Test (A) | Test (M) | Test (C) |
|---|---|---|---|---|---|---|---|
| Scene | Camera | | | | | | |
| Set1:Natural illuminations | iPhone 14ProMax | 55 | 5 | 17 | - | 4 | 7 |
| | Xiaomi11 Pro | 40 | 5 | 19 | - | - | 2 |
| Set2:Artificial illuminations | Huawei Mate30 | 50 | 5 | 10 | 10 | - | 26 |
| | Vivo iqooneo5 | 36 | 6 | 8 | 6 | - | 13 |
| Set3:Mixed illuminations | Huawei P30Pro | 151 | 30 | 38 | 18 | 7 | 78 |
| | Xiaomi 13 | - | - | 8 | 4 | 2 | 2 |

Table 1: The sequence split of our dataset in different dimensions. Note that the cameras within the same set share similar scenes. (N: natural illuminations, A: artificial illuminations, M: mixed illuminations, C: complex illuminations)

where $Q$, $K$, $V$ are computed as,

$$Q = W^Q v_i, K = W^K[v_{i-1}, v_{i+1}], V = W^V[v_{i-1}, v_{i+1}], \quad (5)$$

where $W^Q$, $W^K$, $W^V$ are linear convolutions for feature projection.

Intuitively, cross-frame attention identifies and combines similar spatial features from preceding frames to the target frame, to gather its spatial-temporal features. This implies that the illumination of frame $I_t$ is influenced by information common to different frames to ensure temporal continuity. As in Fig.3, when facing the input frames with content changed (Fig.3(a)), the temporal features extracted by our CTANet (red scatters in Fig.3(f)), is close to these of preceding frames (other scatters), thus improving the accuracy (Average) and stability (STD) of illumination estimation, as in Fig.3(d).

## 5 EXPERIMENTS

In this section, we leverage the BCC dataset and CTA dataset to provide an indepth benchmark analysis of state-of-the-art AWB and TAWB baseline methods as well as our CTANet.

### 5.1 Experiment setting

*5.1.1 Compared methods.* predicting a fixed illumination color for input sequences. We select six AWB methods that are classic and have been compared in several papers (Gray World[24], White Patch[24], Shades of Edge[24], FC4[14]), or are SOTA methods that can be tested on unknown cameras (Quasi[2] and TLCC[23]). Two available TAWB methods (RCCNet[19] and TCCNet[18]) are also chosen for comparison.

*5.1.2 Dataset Splits.* We have strategically split our CTA dataset to address specific challenges related to illuminations and cameras in the TAWB task. Understanding the impact of these two factors on TAWB methods is crucial for achieving targeted improvements. As in Table.1, our CTA dataset was divided into three groups: ***CTA-Set1*** focused on natural illuminations, ***CTA-Set2*** on artificial illuminations, and ***CTA-Set3*** on mixed illuminations (transitions of scenes with different type). To ensure consistent evaluation across various cameras in each group, we meticulously selected similar scenes for each camera for training, validation, and testing. We have also subjected TAWB methods to tests under mixed (transitions of scenes with the same type) and complex (dark and multiple) illuminations in each group. This approach not only assesses the robustness of AWB methods under varying illumination conditions but also evaluates their response to specific camera characteristics,

| Type | Methods | Spatial AE↓ | Temporal MIC↓ | Temporal STD↓ |
|---|---|---|---|---|
| Static-based AWB | Gray Word[24] | 5.32 | 2.70 | 3.20 |
| | White Patch[24] | 6.98 | 6.31 | 5.07 |
| | Shades of Edge[24] | 5.31 | 4.10 | 3.49 |
| CNN-based AWB | FC4[14] | 3.63 | 4.55 | 2.67 |
| | Quasi[2] | 4.28 | 3.53 | 2.95 |
| | TLCC[23] | 9.53 | 10.58 | 4.90 |
| TAWB | RCCNet[19] | 2.48 | 5.44 | 2.71 |
| | TCCNet[18] | 2.54 | 5.19 | 2.66 |
| | CTANet w/o *ST att* | 2.43 | 4.55 | 2.24 |
| | CTANet with *ST att* | **2.38** | **4.40** | **2.16** |

Table 2: Evaluations in the BCC dataset for different methods. The best results are bolded (A lower value of AE means a better illumination estimation correction performance and lower values of MIC and STD mean better stability performance, ST Att: ST Attention in ST-Stage block).

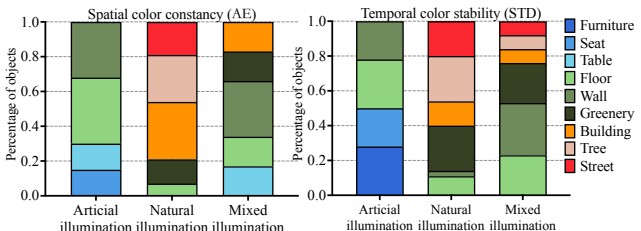

Figure 8: Scene content statistics for the worst 10% test sequences.

fostering the development of TAWB methods that are versatile and comprehensive.

*5.1.3 Evaluation Metrics.* Our evaluation focused on two main goals of TAWB: accuracy and stability of illumination estimation. First, we used the established angular error (AE) [12], which has a lower value indicating more accurate illumination estimation performance. Then, we calculated the MIC (maximum illumination change) and STD (the standard deviation of temporal illumination distribution in ARC diagram[6]), which have lower values means a more temporal stable performance[8]. More details are in the Appendix.

*5.1.4 Implementation Details.* The training of our CTANet is implemented on Pytorch. We apply random cropping, horizontal flipping, rotation, and color jitter for data augmentation. The learning rate is set to $3 \times 10^{-4}$, and the network is trained by Adam optimizer with default parameters for 200 epochs. The training details of other methods are followed by their papers.

### 5.2 Experiment Results on BCC

The quatantiatative results of AWB and TAWB methods on BCC dataset are summarized in Table.2.

For AWB methods, each frame in the sequence serves as input, utilizing a pre-trained model for evaluating. In contrast, for TAWB

| Type | Methods | Test (Natural illuminations) | | | Test (Artificial illuminations) | | | Test (Mixed illuminations) | | | Para meters (M) | Inference time (s) |
|---|---|---|---|---|---|---|---|---|---|---|---|---|
| | | AE↓ | MIC↓ | STD↓ | AE↓ | MIC↓ | STD↓ | AE↓ | MIC↓ | STD↓ | | |
| Static-based AWB | Gray Word [24] | 5.80 | 1.62 | 0.48 | 8.60 | 0.57 | 0.71 | 8.30 | 2.41 | 1.24 | - | 0.0015 |
| | White Patch [24] | 3.84 | 1.82 | 1.12 | 4.12 | 1.35 | 1.03 | 3.28 | 2.80 | 1.90 | - | 0.0010 |
| | Shades of Edge [24] | 4.62 | 0.10 | 0.03 | 14.77 | 0.16 | 0.13 | 14.91 | 0.35 | 0.16 | - | 0.0014 |
| CNN-based AWB | FC4 [14] | 4.01 | 1.70 | 0.84 | 4.94 | 1.59 | 0.82 | 2.99 | 1.53 | 0.73 | 6.52 | 0.0032 |
| | Quasi [2] | 5.46 | 2.84 | 1.68 | 13.45 | 1.12 | 1.75 | 16.15 | 1.96 | 0.94 | 622.87 | 0.0605 |
| | TLCC [23] | 4.75 | 6.00 | 2.11 | 2.64 | 4.20 | 1.77 | 4.61 | 3.99 | 1.72 | 125.71 | 0.0111 |
| TAWB | RCCNet [19] | 2.28 | 2.40 | 1.97 | 1.85 | 0.89 | 0.87 | 2.76 | 1.31 | 1.02 | 20.42 | 0.0156 |
| | TCCNet [18] | 3.36 | 2.86 | 1.88 | 2.59 | 0.83 | 0.88 | 2.83 | 1.38 | 1.00 | 68.80 | 0.0400 |
| | CTANet w/o *ST Att* | 2.49 | 2.78 | 1.82 | 1.80 | 1.02 | 0.89 | 2.81 | 1.50 | 1.10 | **4.43** | **0.0029** |
| | CTANet w/ *ST Att* | **2.26** | **2.32** | **1.80** | **1.71** | **0.78** | **0.86** | **2.73** | **1.27** | **0.99** | 6.37 | 0.0031 |

**Table 3: Evaluations in the CTA dataset for different illumination types (train and test on the same camera). The best results are bolded. (A lower value of AE means a better illumination estimation correction performance and lower values of MIC and STD mean better stability performance, ST Att: ST Attention in ST-Stage block).**

methods, we incorporate the target frame along with its adjacent frames (totaling 3 frames) for prediction during training. In terms of illumination estimation accuracy, TAWB methods outperform AWB methods, due to the beneficial features provided by adjacent frames for predicting the current frame's illumination. However, in temporal consistency, RCCNet and TCCNet underperform compared to some AWB methods (FC4 and Quasi). This difference may be due to the fixed-parameter gating mechanism of LSTM, which makes it difficult for RCCNet and TCCNet to adapt to sudden shifts in content between frames. Benefiting from the self-adjustment to variations in content and illumination, our CTANet achieves the best performance on the accuracy and stability of illumination estimation.

## 5.3 Experiment Results on CTA

*5.3.1 Experiments on Different Illumination Types.* The quantitative results of AWB and TAWB methods for different illumination types are in Table.3. Note that the cameras are kept the same in each test. We trained the methods separately on datasets containing natural illuminations (CTA-Set1), artificial illuminations (CTA-Set2), and mixed illuminations (CTA-Set3). Subsequently, we tested these models on corresponding scenes to evaluate their performance.

**Overall analysis.** Under three typical illuminations, TAWB methods often exhibit lower AE values compared to AWB methods. This suggests that TAWB methods are more effective in illumination estimation accuracy. However, it's observed that some AWB methods, despite having higher AE, may still show lower MIC and STD. Additionally, our CTANet achieved the lowest values in AE, MIC, and STD with the minimum number of parameters, demonstrating its effectiveness and efficiency in addressing both spatial and temporal aspects of illumination.

**Different illumination analysis.** For most methods, their AE for mixed illumination tends to be higher than that for natural and artificial illumination. MIC and STD for natural illumination appear higher than for artificial illumination and mixed illumination. This difference may arise from the complex color temperature in outdoor scenes and the homogeneous sequences in artificial settings. Mixed sequences fall in between. Our CTANet outperformed other methods across all three sets.

**Scene content statistics for 10% worst test sequences.** To especially study the limitations of TAWB methods under different

illuminations, we counted the intersection of sequences from the worst 10% of results across different illumination scenes, using the content annotations from the CTA dataset. In Fig.8, the scene content statistics exhibit a consistent pattern under AE and STD metrics. Predominantly, these scenes contain single-colored objects with minimal edge information. For instance, in indoor scenes with artificial illumination, walls, floors, and seats form the majority of objects. In contrast, outdoor scenes with natural illumination are mostly composed of trees, streets, buildings, and green areas. In transitional scenes with mixed illumination, a significant presence of large walls, floors, and greenery is observed. Such challenges are due to the less informative nature of the above statistical objects, which hampers effective network inference.

**Summary.** Overall, in terms of illumination estimation accuracy, RCCNet, TCCNet and our CTANet achieved more comparable performance than AWB methods under three typical illuminations. Our CTANet addresses this by only 6.37M parameters, which are 31.20% for RCCNet, and 9.25% for TCCNet, as well as faster than RCCNet and TCCNet about 5.3 times and 12.9 times. Notably, solid color scenes with large areas of the same color and fewer edges are one of the challenges of TAWB methods. The illumination estimation stability challenge is indicated in the natural illumination scenes, with larger MIC and STD values of RCCNet and TCCNet. As per the analysis in Sec.2.2, this challenge can be mitigated by adapting to content and illumination changes in our CTANet. Some visual comparisons are in Fig.9, and more comparative effects are in the Appendix. Comparisons of visual effects are shown in Fig.9, where we can see that CTANet has the best and most stable correction in natural light, artificial light and other scenes. *More results are in the Appendix.*

*5.3.2 Experiments on Different Cameras.* Due to the lack of cameras, there has been no benchmarking of the cross-camera effects of the TAWB methods to date. We have supplemented this gap by utilizing data collected from six cameras. Specifically, for camera groups, we trained three TAWB methods on the sequences captured by Iphone14 promax, Huawei Mate30, Huawei P30Pro, and tested in those capture by Xiaomi11 pro, Vivo iqoneo5, Xiaomi13 separately. The test results of inter-camera are in Table.4. In terms of illumination estimation accuracy, cross-camera tests for RCCNet, TCCNet, and CTANet (w/o and w/ *ST att*) methods demonstrate

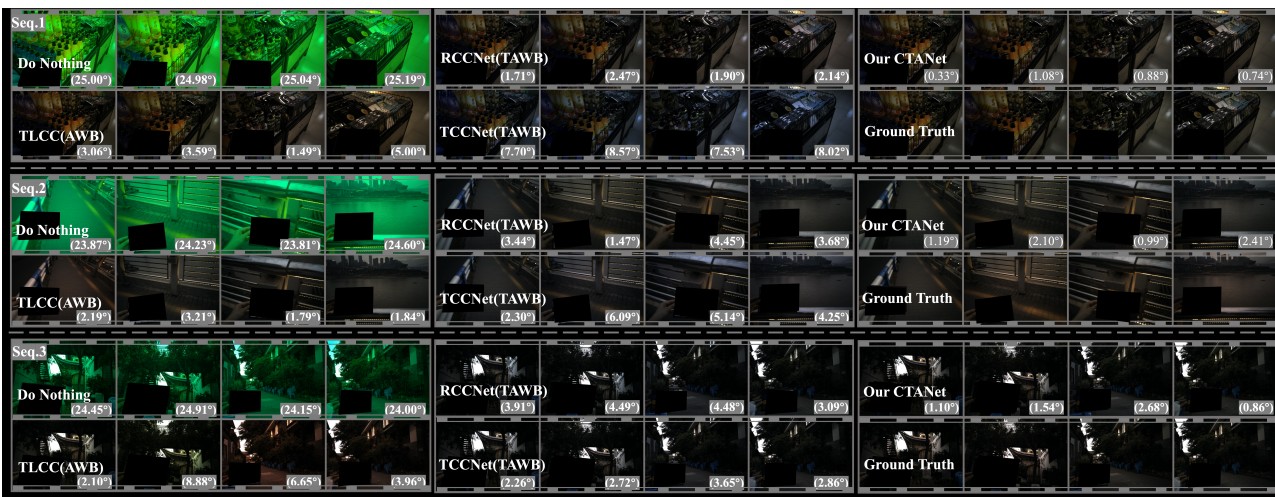

**Figure 9: Color-corrected sequence examples and their angular errors (More visuals in the Appendix).**

| Methods | | Iphone14 promax-Xiaomi11 Pro | Huawei Mate30-Vivo iqooneo5 | Huawei P30Pro-Xiaomi13 |
|---|---|---|---|---|
| RCCNet[19] | AE↓ | 4.53 | 7.81 | 6.73 |
| | MIC↓ | 1.37 | 1.77 | 1.42 |
| | STD↓ | 0.78 | 1.12 | **0.63** |
| TCCNet[18] | AE↓ | **4.26** | 8.16 | 7.39 |
| | MIC↓ | 1.48 | **1.20** | 1.43 |
| | STD↓ | 0.75 | 1.15 | 0.75 |
| CTANet w/o *ST Att* | AE↓ | 4.35 | **7.35** | 6.66 |
| | MIC↓ | 1.58 | 1.36 | 1.44 |
| | STD↓ | 0.42 | 0.96 | 0.69 |
| CTANet w/ *ST Att* | AE↓ | **4.26** | 7.34 | **6.53** |
| | MIC↓ | **1.32** | 1.21 | **1.40** |
| | STD↓ | **0.36** | **0.93** | 0.67 |

**Table 4: Evaluations in the CTA dataset for intra-cameras. The best results are bolded.**

unsatisfactory performances (high AE) compared to the tests in Table.3. This disparity is attributed to the inherent differences between cameras, a key challenge that the TAWB task has not yet adequately addressed. It's worth noting that compared to RCCNet and TCCNet, our CTANet (w/o and w/ *ST att*) still performs better in AE, MIC and STD.

Additionally, models trained using data from Huawei devices, specifically the Huawei Mate30 and Huawei P30 Pro, exhibit the poorest performance when applied to other mobile cameras. This underperformance is primarily due to the marked differences in sensor technologies between Huawei models and other mobile cameras. Addressing the challenge of improving the cross-camera effectiveness of existing TAWB methods remains a critical task for future advancements in the field of TAWB.

### 5.4 Ablation Study

We first examine the effectiveness of RepViT (CTANet w/o *ST Att*) through comparing it with CNN-based AWB methods with best performances. As in Table.3, CTANet w/o *ST Att* obtains 37.91% and 6.02% reduction in AE compared with FC4 under natural illuminations and mixed illuminations, 14.69% reduction in AE compared with Quasi under artificial illuminations. However, the illumination estimation stabilization of CTANet w/o *ST Att* are not optimal in

three typical illumination. When ST Attention added into RepViT, *i.e.*, CTANet w/ *ST Att*, AE continues to decrease, and STD and MIC errors are effectively minimized as well, which showed that incorporating cross-frame attention and RepViT can effectively improve the accuracy and stabilization of illumination estimation. In detail, AE, STD and MIC are reduced by 9.24%, 16.55%, 1.1% in natural illumination scenes, 5%, 23.53%, 3.37% in artificial illumination scenes, 2.84%, 15.33%, 1.82% in mixed illuminations. Also, as discussed in Sec.5.3.1, CTANet w/ *ST Att* achieved the best performance among all compared AWB and TAWB methods, which confirms that the proposed elements are effective in promoting the model's performance.

## 6 CONCLUSION

In this paper, we address the key issues in the field of temporal automatic white balance (TAWB) by introducing innovative solutions in datasets, methods, and benchmark. Firstly, a large-scale dataset called **C**omprehensive **T**emporal **A**WB (CTA) was developed to overcome the limitations of the existing dataset, which lacked diversity and dynamic range; it features a broad array of continuous illuminations across multiple cameras, enhancing realism and applicability. Secondly, we introduced CTANet, designed to address the adaptability issues of existing LSTM-based methods. By incorporating cross-frame attention and RepViT, CTANet dynamically adjusts to changes in content and illumination of input frames, significantly outperforming existing methods. Finally, we have constructed the most thorough benchmark, evaluating eight cutting-edge AWB and TAWB methods against CTANet across three illumination scenes and seven cameras on both BCC and CTA datasets.

Despite these advancements, TAWB research still struggles with temporal color instability in more variable and uncontrolled environments, such as dark illumination or low-information scenes, and the ability to generalize across multiple cameras. We aim to address these challenges in the further work. We hope that our work can contribute to the development of the TAWB field and its integration with other tasks, such as multimodal tasks.

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
