# OpenReview forum: "Thinking Temporal Automatic White Balance: Datasets, Models and Benchmarks"
_acmmm.org/ACMMM/2024/Conference — MM2024 Poster_

### Official Review · Reviewer_iNAP · 2024-05-15

**Rating:** 4
**Confidence:** 3

**Summary:**

The paper focuses on Temporal Automatic White Balance (TAWB) and proposes innovative approaches to improve dataset diversity, model robustness, and benchmarking methodologies. The authors introduce a new dataset called CTA, featuring extensive illumination variations captured with multiple cameras. They also propose a new model, CTANet, integrating cross-frame attention and RepViT to adapt to changes in content and illumination dynamically. A comprehensive benchmark analysis compares their method against existing TAWB and AWB methods across different scenarios and cameras.

**Strengths:**

1.	The introduction of the CTA dataset addresses a gap in TAWB research by incorporating diverse real-world illuminations and multiple camera types, enhancing the realism and applicability of the dataset.
2.	The proposed CTANet model exhibits technical ingenuity, especially in its use of cross-frame attention combined with RepViT, facilitating better adaptation to temporal changes in illumination and content.
3.	The extensive comparative analysis across various methods, illumination types, and camera models provides a robust validation of the proposed model's effectiveness, surpassing existing methods in both accuracy and stability of illumination estimation.

**Limitations:**

1.	While the dataset and model represent significant advancements, the paper does not sufficiently explore or innovate in the underlying theoretical aspects of TAWB, which might limit the depth of contributions to the field's theoretical foundations.
2.	The proposed methods show improved performance on the newly introduced dataset but lack sufficient evidence of their effectiveness across widely diverse real-world applications outside the controlled experimental setups described.
3.	The evaluation heavily relies on the performance metrics that favor the characteristics of the newly introduced dataset and model. This could introduce bias in the benchmarking process, potentially overstating the effectiveness of the proposed methods compared to existing approaches.

**Suitability:**

3

---

### Official Review · Reviewer_6D4i · 2024-05-25

**Rating:** 4
**Confidence:** 2

**Summary:**

This paper conducts a comprehensive analysis of the temporal automatic white balance challenge. It proposes a new CTA dataset to address the lack of dataset issue in this task. Then it further proposes CTANet that integrates cross-frame attention to adapt to varying content and illuminations. In the end, the paper presents a benchmark for this problem

**Strengths:**

This paper provides an extensive analysis of the TAWB challenge and a new dataset for this challenge

This paper proposes CTANet based on ViT, which outperforms prevailing LSTM-based methods

Extensive experiments on the proposed datasets demonstrate the effectiveness of the proposed model and provide a benchmark for the TAWB challenge

**Limitations:**

Although the paper is overall clear, there are some minor writing issues. For example:

1. What is the meaning of the red dotted lines in Figure 2c?

2. In lines 441 and 443, the Figure reference is inconsistent. (Fig. 4a and Fig. 6(a))

3. Lines 376 and 303. Need space before brackets

**Suitability:**

3

---

### Official Review · Reviewer_xQ8P · 2024-06-03

**Rating:** 6
**Confidence:** 3

**Summary:**

To advance TAWB research, this paper addresses challenges in datasets, models, and benchmarks by creating a comprehensive dataset (CTA) with 12K continuous illuminations, proposing CTANet for self-adjustment to content and illumination variations, and establishing a benchmark comparing 8 methods across various scenes and cameras.

**Strengths:**

1. This provide good contribution for Temporal Automatic White Balance in three aspects: datasets, models, and benchmarks.
2. Comprehensive experiments showcase the superiority of the proposed method, dataset, and benchmark.
3. The submitted code ensures reproducibility of the research.
4. From my perspective, the dataset and benchmark contributions are more impactful than the method, greatly accelerating research progress in this field.

**Limitations:**

I see no major drawbacks in this submission. While the proposed method may appear somewhat supplementary and lacks the novelty compared to the dataset and benchmark, it still offers a solid baseline for future research.

**Suitability:**

3

---

### Meta-Review · Area_Chair_Z9Xw · 2024-07-04

**Recommendation:** Accept (Poster)
**Confidence:** 5

**Metareview:**

All reviewers believe this paper delivers solid contributions that should be communicated at MM24. Although there is minor concern on the theoretical insights, the dataset and the benchmark are believed useful. Therefore, AC recommends clear acceptance. Due to limited scope of technical innovations and audience's interest in general, a poster presentation is recommended.